# LST: Ladder Side-Tuning for
# Parameter and Memory Efficient Transfer Learning

**Yi-Lin Sung**    **Jaemin Cho**    **Mohit Bansal**
UNC Chapel Hill
{ylsung, jmincho, mbansal}@cs.unc.edu

## Abstract

Fine-tuning large pre-trained models on downstream tasks has been adopted in a variety of domains recently. However, it is costly to update the entire parameter set of large pre-trained models. Although recently proposed parameter-efficient transfer learning (PETL) techniques allow updating a small subset of parameters (e.g. only using $2\%$ of parameters) inside a pre-trained backbone network for a new task, they only reduce the training memory requirement by up to $30\%$. This is because the gradient computation for the trainable parameters still requires backpropagation through the large pre-trained backbone model. To address this, we propose Ladder Side-Tuning (LST), a new PETL technique that can reduce training memory requirements by more substantial amounts. Unlike existing parameter-efficient methods that insert additional parameters inside backbone networks, we train a ladder side network, a small and separate network that takes intermediate activations as input via shortcut connections (called *ladders*) from backbone networks and makes predictions. LST has significantly lower memory requirements than previous methods, because it does not require backpropagation through the backbone network, but instead only through the side network and ladder connections. We evaluate our method with various models (T5 and CLIP-T5) on both natural language processing (GLUE) and vision-and-language (VQA, GQA, NLVR[2], MSCOCO) tasks. LST saves $69\%$ of the memory costs to fine-tune the whole network, while other methods only save $26\%$ of that in similar parameter usages (hence, 2.7x more memory savings). Moreover, LST achieves higher accuracy than Adapter and LoRA in a low-memory regime. To further show the advantage of this better memory efficiency, we also apply LST to larger T5 models (T5-large, T5-3B), attaining better GLUE performance than full fine-tuning and other PETL methods. The trend also holds in the experiments on vision-and-language tasks, where LST achieves similar accuracy to other PETL methods when training a similar number of parameters while also having 2.7x more memory savings.[1]

## 1    Introduction

Recently, large-scale pre-training and fine-tuning of transformers [54] have been successful in various domains [8, 36, 44, 38, 53, 7, 20, 1, 10]. As the model size grows rapidly, fine-tuning the entire parameter set of the large pre-trained model has become very costly. Parameter-efficient transfer learning (PETL) [31, 23, 39, 40, 34, 51, 58, 17, 24, 18, 42, 60, 14, 52, 59] is a recent research direction for online or multi-task learning. The goal is to build a system that performs well on all tasks without training an entire new model for every new task. Concretely, PETL methods select a small subset of pre-trained parameters and/or insert a few parameters to a pre-trained network and update those parameters for new tasks, while freezing most of the original parameters. In the

---

[1]Our code is available at: https://github.com/ylsung/Ladder-Side-Tuning.

36th Conference on Neural Information Processing Systems (NeurIPS 2022).

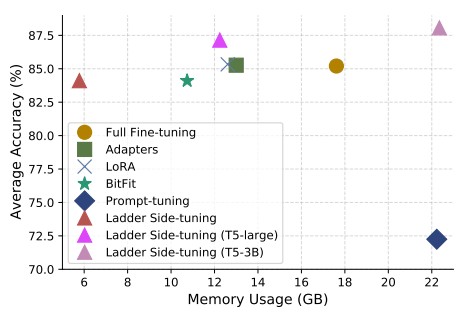

Figure 1: Comparison between full fine-tuning, Adapter, LoRA, BitFit, and Ladder Side-Tuning over GLUE tasks. The y-axis is the average accuracy of 8 GLUE tasks, while the x-axis is the GPU memory usage during training. Unless specially stated, we use the T5-base in the figure.

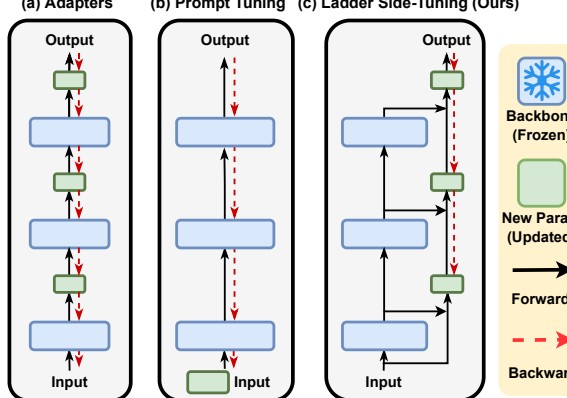

Figure 2: Comparison between transfer learning with (a) Adapters, (b) Prompt Tuning, and our (b) Ladder Side-Tuning (LST). LST reduces memory usage by removing the need of backpropgation through backbone networks.

natural language processing (NLP), computer vision (CV), and vision-and-language (VL) domains, two types of parameters have been commonly updated for parameter-efficient transfer learning: (a) *Adapters* [23, 40, 39]: small modules inserted into transformer blocks; (b) *Prompt* [34, 31]: small parameters concatenated with input embeddings (see Figure 2).

However, while parameter-efficient techniques reduce the number of parameters to update, they do not reduce the memory requirement during training by much (up to 30%). In other words, if a large pre-trained language model does not fit on a GPU, these techniques usually do not help to fit the model on the GPU. In other words, these techniques usually do not help to fit a large pre-trained language model on a GPU if the model cannot be trained on the GPU with standard fine-tuning. Since the updated parameters are inside the backbone language models, to calculate gradients for these parameters for backpropagation, we still need to run the backward pass through the large pre-trained language models. This prevents PETL methods from being applied to many real-world applications with limited computational resources.

To address this issue, we propose Ladder Side-Tuning (LST), a memory-efficient PETL method. LST separates trainable parameters from the backbone model to construct a side network, which is responsible for adapting the entire model to new tasks. Concretely, we train a *ladder* side network, a lightweight network that takes intermediate activations via shortcut connections (ladders) from the backbone networks as input and makes predictions. As shown in Figure 2, unlike previous (a) adapters and (b) prompt tuning methods, (c) our LST does not add trainable parameters inside the pre-trained model, and this completely eliminates the need for expensive backpropagation of a large backbone network and saves substantial memory during transfer learning. Instead of initializing the side network's weights randomly, we utilize a network pruning technique to retrieve a smaller pruned network and use it as the side network. In addition to our standard design of the side network, we also boost the efficiency of LST by dropping layers of the side network. We empirically demonstrate that the layer dropping can significantly improve both memory and parameter efficiency without sacrificing performance. Furthermore, during inference, even though we have to propagate forward through two distinct networks, LST does not necessarily use more inference time because the same level of the backbone network and the side network can be computed in parallel.

We conduct comprehensive studies on LST using diverse NLP and VL tasks, namely, GLUE [55], VQA [16], GQA [25], NLVR² [50] and MSCOCO [6]. Overall, in GLUE experiments, LST saves 69% of the GPU memory that is needed for fine-tuning the entire backbone model, saving 2.7x memory compared against Adapter and LoRA. Also, LST achieves higher accuracy than other PETL methods in a low-memory regime. To take advantage of this better memory efficiency, we also apply LST to larger language models (T5-large and T5-3B) and find that it achieves higher accuracy than other PETL techniques when GPU memory utilization is similar. The findings still hold in the Vision-Language (VL) experiments; LST is not only a method that can fit in a 16GB GPU with 7.5% trainable parameters, but it also has similar or better accuracy than other PETL methods (and

again with 2.7x memory savings). To justify our design of LST, we conduct ablation studies on initialization strategies and alternatives to add shortcut connections. The results reveal that these components considerably help performance with minor computation overhead.

## 2 Related Work

### 2.1 Parameter-efficient Transfer Learning (PETL)

**PETL for NLP.** In the past few years, large pre-trained language models have made huge success in NLP. Parameter-efficient transfer learning (PETL) is a research direction that reduces computational cost of adapting large pre-trained models to new tasks, by avoiding updates of the entire parameters. A popular line of work on PETL is to add a few trainable parameters and only tune them. For example, Adapters [48, 23] are small bottleneck modules that are inserted into transformer layers, and experiments have shown that training adapters with layer normalization layers is sufficient to achieve full fine-tuning performance. In a similar trend, LoRA [24] injects trainable rank decomposition matrices into a frozen pre-trained model. Instead of inserting new parameters into pre-trained models, prompt-based [31, 34] methods add trainable parameters to the input and keep the entire pre-trained model unchanged during training. The inserted prompts learn to make use of the knowledge of the pre-trained model to solve new tasks. Although the concept of adapter-based and prompt-based approaches is different, He et al. [18] unify the two lines of approaches (including LoRA) into adapter-based methods. In addition to approaches that introduce new parameters, there are also various methods [51, 58, 17] that select a sparse subset of parameters from the pre-trained model to update, without adding any new parameters. One of such representative methods is BitFit [58], which updates every bias term in the model.

**PETL for CV/VL.** While most of the progress in PETL is made in NLP domain, researchers have also applied this technique to the CV [47, 46, 27, 59, 62, 28, 60, 21] and VL [52, 61, 62, 28, 60] domains. VL-Adapter [52] benchmarks adapter-based and prompt-based methods on multiple image-text and video-text tasks, and shows adapters enable us to efficiently learn fusion information of vision and language. On the CV side, benefitting from the parameter efficiency of adapters and prompt-tuning, some works [62, 28, 60] apply these approaches to CLIP [43] to achieve strong few-shot performance in image classification tasks. Side-Tuning [59] uses an additive side network, which sums its representation with the backbone network in the last layer, to solve various tasks with ResNet [19] and BERT [8]. Although LST takes inspiration and has similarities to Side-Tuning, we argue that there are major differences in motivations, architecture designs, and applied tasks between the two methods. LST aims to reduce the memory requirement of current PETL methods, whereas Side-Tuning does not focus on memory reduction (sometimes their side network is even as big as the backbone network), but instead their motivation is to ease the forgetfulness in incremental learning. Our ladder side network is more robust than their design because the shortcuts fuse the intermediate information from the backbone network, and we also use layer dropping and network pruning techniques to make LST more efficient and stronger. Lastly, we further extend LST in VL architecture and demonstrate its usefulness on multiple VL tasks.

Current PETL approaches explore how to achieve competitive results using as few parameters as possible. However, parameter efficiency does not necessarily mean memory efficiency. In this work, we propose LST that has these two benefits simultaneously. Concurrently, Liu et al. [37] also propose Y-tuning to address a similar issue; it exhausts all possible labels and feeds them into a model to select the best answer from the input. However, it is intractable oftentimes to list all answers in some tasks, for example, regression and open-ended generation tasks. On the other hand, LST is more flexible in applying to different architectures and tasks. We show that LST can outperform Y-tuning with fewer parameter updates in Table 2.

### 2.2 Memory-Efficient Training

Memory-efficient training aims to reduce the memory cost when training neural networks. Some approaches achieve this goal by cutting down the storage of intermediate activations, which dominate the training cost, to release a large amount of memory. The design of reversible neural networks [15, 29, 41] allows the model not to save intermediate activations because each layer's activations can be reconstructed from the next layer's. Gradient checkpointing [5] proposes to trade computation for memory by dropping some intermediate activations and recovering them from an extra forward

pass. LST uses a different approach to cut the storage of activations; it keeps the backbone model frozen and constructs a side network for cheaper training. Since the backbone model is not updated, LST is not only memory-efficient but also parameter-efficient. Furthermore, memory saving from reversible neural networks and checkpointing is agnostic to memory saving from LST. Researchers can combine those methods with LST if they pursue a higher level of memory efficiency.

Another line of memory-efficient methods is network compression, which compresses a backbone network to a smaller one, and the generated network is cheaper for both training and inference. Two popular approaches for compression are network pruning and distillation. Network distillation [22, 30] constructs a student network and force it to have same output distribution of the teacher network over a chosen dataset. Network pruning [12, 13] makes models lighter by learning importance scores of parameters and trimming unimportant parameters or neurons. While PETL still uses those untrained parameters in the forward pass, network compression entirely discards them or sets them to zero. As a result, network compression can generate models for faster inference speed while PETL can achieve better performance by updating fewer parameters.

In this paper, we explore using network pruning or network distillation [22] (used in Side-tuning) to extract a sub-network that contains critical information of the backbone model and use it for initializing our side network. For distillation, we did not follow the original Side-tuning to apply distillation with large-scale pre-training datasets (e.g., C4 [44]) because it makes distillation hard to conduct with limited resources and ultimately violates our goal of "efficient training." Also, using an extra pre-training dataset during fine-tuning is unfair to other approaches. We use the standard distillation procedure with the T5 [44] pre-training objective to train the side network. That is, the student (side) network learns to predict the masked spans and match the output distribution of the teacher (backbone) network simultaneously. We show the comparison between distillation-based and pruning-based initializations in Figure 8.

We primarily use the network pruning method proposed by Li et al. [33] to initialize the side network because of its efficiency, and we describe the approach in detail in Section 3.3. The standard procedure of network pruning is (1) learn [12, 13] or heuristically define [33] an "importance measure" to identify the importance of parameters, (2) prune $p\%$ of parameters with lower importance scores, (3) repeat the first and second steps until reaching the target sparsity. The rewinding procedure enables pruning techniques to find a more sparse sub-network. In this paper, to keep the whole pruning process efficient, we either use weights magnitude [33] or Fisher Information [51, 35] as importance measures, and reach the target sparsity in one shot. As a PETL method, LST makes use of intermediate information from the backbone model as the inputs, and we empirically demonstrate those additional inputs significantly improve performance in Figure 8.

## 3 Ladder Side-Tuning (LST)

We introduce Ladder Side-Tuning (LST), a new PETL technique that can also reduce training memory requirements by substantial amounts than previous methods. In Section 3.1, we analyze the computational cost for fine-tuning with trainable modules in backbone models. Then we explain the architectural details (Section 3.2), structural weight initialization based on network pruning (Section 3.3), and dropping side network layers for more efficiency (Section 3.4).

### 3.1 Dependency on Backpropagation through Large Backbone Model

We consider a $N$ multilayer perceptron (MLP): $f_N(f_{N-1}(...f_2(f_1(x))...))$, where the $i^{th}$ layer $f_i(x) = \sigma_i(W_i x + b_i)$ consists of weight $W_i$, bias $b_i$, and nonlinear function $\sigma_i$. We denote the output of $i^{th}$ layer as $a_{i+1}$ and the pre-activation as $z_{i+1}$, where $a_{i+1} = \sigma_i(z_{i+1}) = \sigma_i(W_i a_i + b_i)$. In backpropagation with loss $L$, the gradient with respect to $W_i$ and $b_i$:

$$\frac{\partial L}{dW_i} = \frac{\partial L}{\partial a_{i+1}} \frac{\partial a_{i+1}}{\partial z_{i+1}} \frac{\partial z_{i+1}}{\partial W_i} = \frac{\partial L}{\partial a_{i+1}} \sigma'_i a_i, \qquad \frac{\partial L}{db_i} = \frac{\partial L}{\partial a_{i+1}} \sigma'_i \tag{1}$$

where $\sigma'_i$ is the derivative of $\sigma_i$. $\frac{\partial L}{\partial a_{i+1}}$, the gradient with respect to $a_i$, can be calculated with the gradients with respect to $a_{i+2}$, using the chain rule:

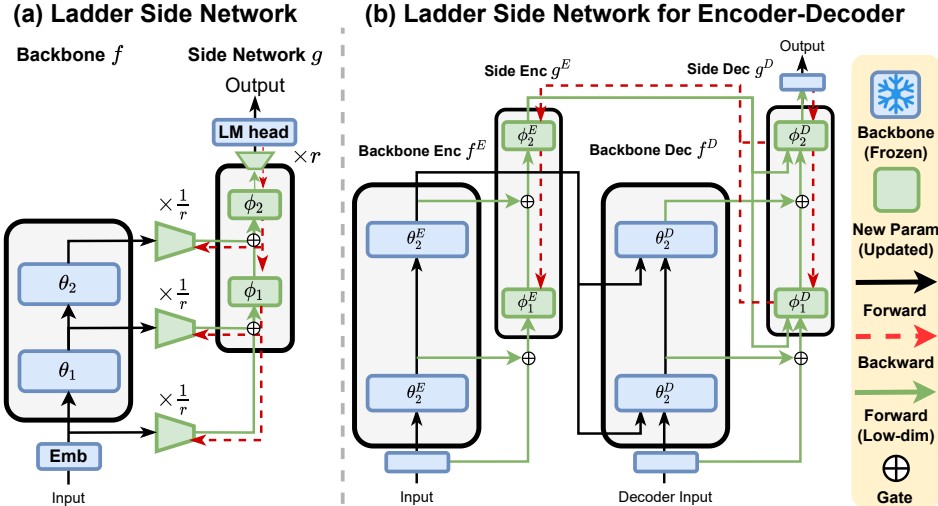

Figure 3: Illustration of Ladder Side-Tuning (LST) with transformers described in Section 3.2. (a) shows a high-level overview of LST, and (b) shows LST with an encoder-decoder architecture.

$$\frac{\partial L}{\partial a_{i+1}} = \frac{\partial L}{\partial a_{i+2}} \frac{\partial a_{i+2}}{\partial z_{i+2}} \frac{\partial z_{i+2}}{\partial a_{i+1}} = \frac{\partial L}{\partial a_{i+2}} \sigma'_{i+1} W_{i+1} \tag{2}$$

As shown in Equations (1) and (2), during backpropagation, there are two terms dominating the memory footprint: 1) $\{a\}$ corresponding to updated parameters $\{W\}$ and 2) $\{\sigma'\}$ that must be cached for the chain rule. Note that we use $\{\cdot\}$ to denote a set of activations, parameters, or gradients. Existing PETL methods, such as Adapters [23], LoRA [24], Prompt-tuning [31], and BitFit [58, 3] could reduce the memory footprint by making $|a|$ smaller, as they have fewer $\{W\}$ to update, but do not reduce $|\sigma'|$, where $|\cdot|$ means the size of set $\{\cdot\}$. Since most activation functions do not change dimensions (i.e., $|a| = |\sigma'|$), the memory footprint for backpropagation $|a| + |\sigma'|$ can be reduced by up to 50% by the PETL methods when they reduce the entire memory footprint for $|a|$. By making the updated parameter do not require backpropagation through the backbone network, our LST can achieve better memory efficiency beyond 50%, and we explain it below.

## 3.2 Ladder Side Network for Transformers

Unlike existing transfer learning methods that insert additional parameters inside a transformer network, we propose training a *ladder* side network, a small and separate network that takes intermediate activations from the backbone transformer as input and makes predictions. As illustrated in Figure 3 (a), since the ladder side network parameters $\phi$ are not used during the forward pass of the backbone transformer with parameters $\theta$, the update of the ladder side network does not require expensive backpropagation of the large backbone transformer. Note that our LST method is not limited to a specific architecture. We provide a simplified overview of LST with an encoder architecture in Figure 3 (a) and an illustration of LST with an encoder-decoder architecture in Figure 3 (b).

**Lightweight architecture.** Our side network $g$ is a lightweight version of the backbone transformer $f$, where all weights and hidden state dimensions of in $g$ are $\frac{1}{r}$ times of the original weights and hidden states of $f$, where $r$ is a reduction factor (e.g. $r = 2, 4, 8, 16$). For example, if the backbone $f$ has a 768-dimensional hidden state, then the side network $g$ with $r = 16$ has a hidden state of 48 dimensions (= 768/16). The side network $g$ reuses frozen word embeddings ('Emb' in Figure 3 (a)) and the language model head ('LM head' in Figure 3 (a)) of the backbone $f$. Following the analysis in Section 3.1, we also examine the memory cost of LST. Recall that original memory footprint for backpropagation is $|a| + |\sigma'|$. Because we do not have to run a backward pass through the backbone network, we can only consider the gradients for the side network, whose memory footprint is $\frac{|a|+|\sigma'|}{r}$. Therefore, LST has a better memory efficiency than other PETL methods (saving up to 50%) as long as $r$ is greater than 2 (we find 8 works well in most experiments).

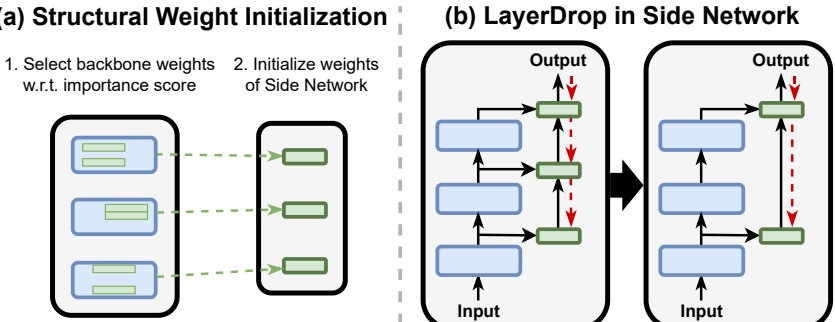

Figure 4: Illustration of (a) Structural Weight Initialization (Section 3.3) and (b) Layer Dropping (Section 3.4). In our experiments, we find that initialization of side network parameters from backbone network parameters improves performance, and dropping some shortcut connections improves efficiency without hurting performance.

**Gated ladder connections.** Although Zhang et al. [59] found that late fusion to combine the representations of the backbone and the side network works well with convolutional networks for CV tasks, in our experiments, we find that late fusion hurts the performance of the transformer architecture in NLP tasks (see Figure 8 in Section 5 for details). To address this, we use the shortcut connection (called *ladder*, due to the overall shape created from the multiple shortcut connections) from intermediate activations from the backbone $f$ to the side network $g$ and find it helpful. We learn linear projections to downsample ($\times \frac{1}{r}$) the intermediate activations (including word embeddings) of $f$ to low-dimensional attention blocks in $g$. Then, we learn a linear projection to upsample ($\times r$) the side network output to the dimension of the original language model head. The linear projections are illustrated as green trapezoids in Figure 3 (a). The $i^{th}$ transformer layer of the side network $g$ combines the activation of the backbone $h_i^f$ and the activation of the previous layer of the side network $h_{i-1}^g$ with learned gating: $\mu_i * h_i^f + (1 - \mu_i) * h_{i-1}^g$, where $\mu_i = \texttt{sigmoid}(\frac{\alpha_i}{T})$ is a gate parameterized with a learnable zero-initialized scalar $\alpha_i$ and temperature $T (= 0.1)$. We have also tried to use Adapter blocks to build the side network and replace the gating mechanism with cross-attentions, but we find the current design works the best (see **??**).

### 3.3 Structural Weight Initialization for Ladder Side Network

We find it helpful to initialize the weights of the side network $\phi$ from the weight of the backbone network $\theta$ based on structural pruning [33], as shown in Figure 4 (a). Concretely, given a weight matrix $W \in \mathbb{R}^{d_{out} \times d_{in}}$ of the backbone network that maps the $d_{in}$-dim vectors to the $d_{out}$-dim space, and the importance matrix of the weight $I \in \mathbb{R}^{d_{out} \times d_{in}}$, we first calculate the importance score of each row $s_i = \sum_j |I_{i,j}|$, denoting the importance of each weight vector. Note that the importance matrix $I$ used in this work are either weight magnitude [33] ($I = W$) or empirical Fisher Information [51] ($I = F_W = \frac{1}{N} \sum_{i=1}^{N} (\nabla_W \log p(y_i|x_i))^2$; $(x_i, y_i), ..., (x_N, y_N)$ are samples from data). Then, we choose the rows of $W$ which have the top $\frac{d_{out}}{r}$ importance scores and prune the remaining rows to obtain a new weight matrix $W^P \in \mathbb{R}^{\frac{d_{out}}{r} \times d_{in}}$. The columns of the weights and the importance matrix in the next layer corresponding to the pruned feature map are also pruned. By iterating this process, we obtain the set of weight matrices whose rows and columns are pruned $\frac{1}{r}$ times from the backbone network and use them to initialize the side network. In our experiments shown in Figure 7, we find that using Fisher information as an importance score metric generally performs well, and therefore we use it in our structural weight initialization.

### 3.4 Layer Dropping in the Ladder Side Network

We explore to increase efficiency of LST even further by making side network more compact, by dropping its intermediate transformer layers, as illustrated in Figure 4 (b). Similar to LayerDrop [11], we drop layers in the side network, and this can linearly reduce the memory and parameter requirements of LST. For instance, a side network with $N$ layers will only have $2^{nd}, 4^{th}, 6^{th} \ldots$ layers left,

after we drop half of the layers. Refer to Section 4 for more details on applying layer dropping on an encoder-decoder architecture. In Figure 6, we show that layer dropping can greatly boost the model's efficiency without sacrificing performance.

## 4    Experiment Setup

**Datasets.**    We evaluate LST on NLP and VL tasks. For NLP tasks, we use the GLUE [55] benchmark, which consists of seven classification and one regression task. The benchmark evaluate models on multiple diverse tasks over linguistic acceptability (CoLA [56]), sentiment analysis (SST-2 [49]), similarity and paraphrase (MRPC [9], QQP [26], STS-B [4]) and natural language inference (MNLI [57], QNLI [45], RTE [2]). For VL tasks, we experiment with visual question answering (VQA [16], GQA [25]), visual reasoning (NLVR$^2$ [50]) and image captioning (MSCOCO [6]) tasks.

**Baselines.**    We compare LST against the full fine-tuning and several popular PETL approaches on both NLP and VL setups. In **Full fine-tuning**, all the parameters are updated for a downstream task. Full fine-tuning is not parameter-efficient nor memory-efficient, but it serves as the upper bound of the fine-tuning performance. To compare to other PETL methods, we reproduce **Adapters**, where we inject small trainable modules after every attention and feed-forward layer, and we solely train those modules and layer normalization layers while keeping the rest of the model frozen. We also reproduce **LoRA**, which inserts trainable low-rank matrices into the model to parameterize the weights' changes. In **BitFit**, we only update the bias terms over the course of training. Lastly, we compare our method to **Prompt-tuning**, where trainable prompt vectors are prepended to the input. We initialize the prompt vectors with the embedding of the pre-trained model's vocabularies.

**Training and Evaluation Setup.**    For NLP tasks, we use T5 [44], a pre-trained encoder-decoder language model as our backbone. We use T5-base in most experiments, except that we scale up LST on T5-large and T5-3B to demonstrate its memory efficiency. The training and evaluation process follows the setup used by Mahabadi et al. [40]. Since there is no local test set, we split 1k samples from the training set as the new validation set and use the original validation set as the test set. For datasets whose samples are less than 10k (RTE, MRPC, STS-B, CoLA), we split the validation set into two equal-sized subsets and treat them as a new validation and test set. For MNLI, we use the mismatched set as the validation set and matched set as the test set. We train every approach with 10 epochs on large datasets and 20 epochs on small ones (RTE, MRPC, STS-B, CoLA) for complete convergence. We search for learning rates over $\{3 \times 10^{-4}, 1 \times 10^{-3}, 3 \times 10^{-3}\}$ for LST and LoRA[24], and we use the optimal learning rates that are used by Mahabadi et al. [40] for other methods. The reduction factor used in LST is set to 8 if not additionally specified. T5-base has 12 layers each in encoder and decoder, while T5-large and T5-3B have 24 layers each. In our experiments, we do not drop layers in T5-base unless we specially mention it. For T5-large and T5-3B, we drop 24 layers (12 layers each in encoder and decoder) and 46 layers (23 each) of the side network to make the memory usage close to our baselines. The experiments on T5 take around 12 hours to train with one A6000 GPU (48GB).

For VL tasks, we experiment with CLIP-T5 [52], which is a VL architecture combining CLIP [43] and T5 [44]. We always freeze the CLIP and only train the T5 for new tasks. The CLIP visual representation is concatenated with the text embedding, and the combined input is fed to T5 to make predictions. A visual projection layer is added between CLIP and T5 to let the visual representation have the same dimension as the text embedding. To avoid updating the visual projection layer by the gradients from the backbone model, we do not feed combined inputs to the backbone model, but only text inputs. The combined inputs are fed to the side network, so we can achieve efficient training by only computing the gradients from the side network. Because the backbone network only uses texts as the input, the information from the backbone network via shortcut connections is only summed to the text part of the side network's combined inputs. We follow the multi-tasking setting for training and evaluation used in VL-Adapter [52]. We report the performance on Karpathy test/test-dev/test-P/Karpathy test split for VQA/GQA/NLVR$^2$/MSCOCO, and train models for 20 epochs. We search learning rates over $\{3 \times 10^{-4}, 1 \times 10^{-3}, 3 \times 10^{-3}\}$ for PETL methods, and use $1 \times 10^{-4}$ used by Sung et al. [52] for full fine-tuning. We set the reduction factor for the side network to 4. We train CLIP-T5 for 16 hours on one A6000 GPU. In **??**, we comprehensively list hyper-parameters for NLP and VL experiments in **??** and **??**, respectively.

Table 1: Comparison between multiple parameter-efficient training methods on GLUE benchmark. We use T5-base if we don't additionally specify. We report accuracy for SST-2, MNLI, QNLI and RTE. For CoLA and STS-B, we use Matthew's Correlation and Pearson-Spearman Correlation as the metrics, respectively. For MRPC and QQP, we report the average of F1 score and accuracy. Each number in the table is the average result over three seeds, and the subscripts are standard deviations. For the results with $^\dagger$, we report the best performance out of three seeds due to the instability of the method. We report the maximum memory usage training and evaluating on RTE for each method.

| Method | Update Param. per Task (%) | Memory Usage (GB) | | CoLA | SST-2 | MRPC | QQP | MNLI | QNLI | RTE | STS-B | Avg. |
|---|---|---|---|---|---|---|---|---|---|---|---|---|
| | | Train | Inference | | | | | | | | | |
| Full fine-tuning | 100 | 17.6 | 0.86 | $62.8_{2.5}$ | $93.9_{0.6}$ | $91.9_{1.0}$ | $89.9_{0.4}$ | $86.2_{0.4}$ | $92.5_{0.3}$ | $74.1_{1.0}$ | $90.3_{0.1}$ | $85.2_{0.4}$ |
| Adapters | 1.63 | 13.0 | 0.87 | $64.4_{1.5}$ | $94.2_{0.5}$ | $88.9_{0.2}$ | $88.9_{0.1}$ | $86.4_{0.2}$ | $93.1_{0.2}$ | $75.1_{0.7}$ | $91.1_{0.2}$ | $85.3_{0.2}$ |
| LoRA | 1.71 | 12.6 | 0.86 | $63.3_{0.1}$ | $94.3_{0.1}$ | $90.1_{0.7}$ | $89.0_{0.1}$ | $86.3_{0.1}$ | $93.2_{0.1}$ | $75.5_{3.3}$ | $90.9_{0.0}$ | $85.3_{0.5}$ |
| BitFit | 0.13 | 10.7 | 0.86 | $61.8_{1.5}$ | $94.3_{0.1}$ | $91.0_{0.2}$ | $88.7_{0.0}$ | $85.6_{0}$ | $93.1_{0.1}$ | $67.6_{0.6}$ | $90.8_{0.2}$ | $84.1_{0.1}$ |
| Prompt-tuning | 0.03 | 22.2 | 0.87 | $0^{\dagger}_{2.5}$ | $90.3^{\dagger}_{16.3}$ | $74.6_{0.0}$ | $88.5_{0.2}$ | $82.5_{0.9}$ | $92.5_{0.2}$ | $59.5_{2.9}$ | $90.1_{0.1}$ | $72.2_{1.6}$ |
| Ladder Side-Tuning | 1.74 | 5.5 | 0.88 | $58.1_{3.2}$ | $94.1_{0.3}$ | $90.4_{1.0}$ | $88.8_{0.1}$ | $85.6_{0.1}$ | $93.3_{0.1}$ | $71.9_{2.1}$ | $90.7_{0.2}$ | $84.1_{0.5}$ |
| Ladder Side-Tuning (T5-large) | 1.23 | 12.2 | 2.88 | $65.3_{1.9}$ | $95.7_{0.1}$ | $91.6_{1.0}$ | $89.7_{0.0}$ | $88.6_{0.0}$ | $94.1_{0.2}$ | $79.9_{0.0}$ | $92.4_{0.1}$ | $87.1_{0.2}$ |
| Ladder Side-Tuning (T5-3B) | 0.08 | 22.4 | 11.01 | $66.4_{1.7}$ | $96.5_{0.1}$ | $92.9_{0.8}$ | $89.7_{0.1}$ | $90.7_{0.1}$ | $95.1_{0.2}$ | $80.1_{1.0}$ | $93.0_{0.3}$ | $88.1_{0.4}$ |

Table 3: Comparison between multiple parameter-efficient training methods on VQA, GQA, NLVR$^2$, and MSCOCO. We use T5-base for all approaches. We report accuracy for VQA, GQA and NLVR while we use CIDEr to evaluate MSCOCO. Each number in the table is the average result over three seeds, and the subscripts are standard deviations.

| Method | Update Param. (%) | Memory Usage (GB) | | VQA | GQA | NLVR$^2$ | MSCOCO | Avg. |
|---|---|---|---|---|---|---|---|---|
| | | Train | Inference | | | | | |
| Full fine-tuning | 100 | 36.2 | 0.86 | $67.1_{0.1}$ | $56.3_{0.3}$ | $74.3_{0.4}$ | $112.2_{0.3}$ | $77.5_{0.3}$ |
| Adapters | 7.98 | 28.4 | 0.93 | $67.1_{0.1}$ | $56.0_{0.4}$ | $72.7_{0.3}$ | $111.8_{0.1}$ | $76.9_{0.2}$ |
| LoRA | 7.54 | 27.9 | 0.86 | $63.7_{0.2}$ | $53.3_{0.1}$ | $70.0_{0.3}$ | $110.3_{0.4}$ | $74.3_{0.1}$ |
| BitFit | 0.83 | 22.7 | 0.86 | $55.1_{0.2}$ | $45.5_{0.2}$ | $51.7_{1.1}$ | $101.2_{0.2}$ | $63.4_{0.1}$ |
| Prompt-tuning | 1.26 | 38.7 | 0.87 | $47.4_{0.7}$ | $40.6_{0.4}$ | $51.0_{0.4}$ | $96.1_{0.9}$ | $58.8_{0.6}$ |
| Ladder Side-Tuning | 7.46 | 15.3 | 0.93 | $66.5_{0.1}$ | $55.9_{0.1}$ | $71.6_{0.3}$ | $113.5_{0.3}$ | $76.9_{0.1}$ |

## 5 Experimental Results

In this section, we show experiments to justify our design of LST and demonstrate that LST performs the best among all approaches in the scenario with limited memory. As the result, LST is the most efficient tool to fine-tune large-scale pre-trained models for real-world applications.

**LST outperforms other methods under similar memory usage.** Figure 1 and Table 1 show the results on GLUE of different approaches applying on T5-base. We drop 6 layers (3 layers each in side encoder and decoder) for LST to match the parameter usage of the Adapter and LoRA. Under the same parameter usage, LST can save 69% of memory cost to fully fine-tune the model, while Adapter and LoRA only save 26% of that, leading to LST having a 2.7x more memory saving. Compared to BitFit, LST achieves the same average performance but costs 5GB less GPU memory. LST also surpasses Prompt-tuning in terms of both performance and memory cost. To further take advantage of the memory efficiency of LST, we also train T5-large and T5-3B with LST. We find that with a similar budget of memory usage in Adapter and LoRA, LST with T5-large can surpass the performance of other methods by a large margin. The result on T5-3B also outperforms the result on T5-large, demonstrating the scalability of our memory-efficiency method on large language models. Furthermore, even though LST increases the model size, its additional inference memory usage is negligible as LST uses almost the same inference memory (0.88 GB) as the full fine-tuning (0.86 GB).

In Table 2, we also compare LST with a concurrent work, Y-tuning [37] on GLUE tasks (except for STS-B) with BART-large [32] encoder as backbone. Following their experimental setup, we use a different learning rate and report the best accuracy out of three seeds for each task. Overall, LST outperforms Y-tuning by a large margin with fewer updated parameters.

Table 2: LST vs. Y-tuning.

| Method | Update Param. per Task (%) | Avg. GLUE |
|---|---|---|
| Y-tuning | 7.7 | 76.9 |
| LST | 2.6 | 82.1 |

**LST is competitive on VL tasks.** As we have mentioned beforehand, we also extend LST on a multi-modal architecture, CLIP-T5, on multiple VL tasks, and we demonstrate the outcome in Table 3.

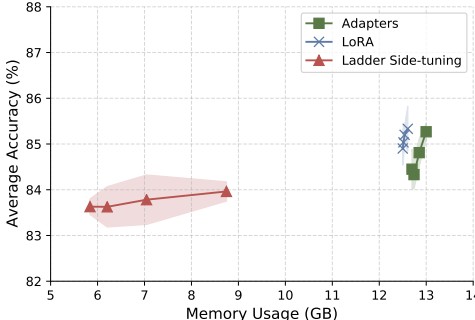

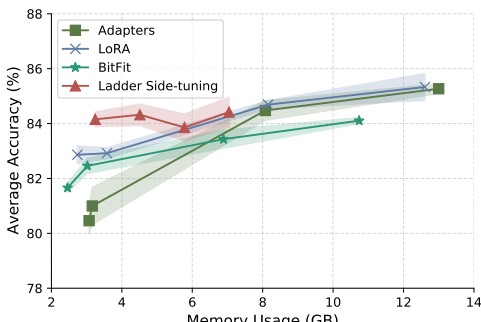

Figure 5: The accuracy-memory trade-off for Adapter, LoRA, and Ladder Side-Tuning over GLUE tasks. We vary the reduction factor in Ladder Side-Tuning, hidden dimension in Adapter, rank in LoRA to get the architectures with different training costs.

Figure 6: The accuracy-memory trade-off for Adapter, LoRA, BitFit, LST over GLUE tasks. we drop $N \in \{0, 6, 12, 18\}$ layers in an inter-leaving manner for LST while we gradually freeze the first $N \in \{0, 6, 12, 18\}$ layers in other methods (also remove inserted parameters in those layers).

With similar parameter usage, LST is the only method that can fit into a single 16GB GPU. Besides the efficiency, it is as competitive as full fine-tuning and Adapter, outperforming other PETL approaches.

**LST performs the best in low-memory regime.** To have a better understand of the memory advantage of LST, we adjust the hyper-parameters in our method (reduction factor $\in \{32, 16, 8, 4\}$), Adapter (hidden dimension $\in \{6, 12, 24, 48\}$) and LoRA (rank $\in \{4, 8, 16, 32\}$) to create multiple architectures with different memory costs. Figure 5 shows the performance and memory efficiency trade-off for all methods. We find the memory saving is not obvious for Adapter and LoRA, because the gradients of the backbone model's intermediate outputs are still computed (see Section 3.1 for details). Even though the Adapter and LoRA can get slightly better memory efficiency by reducing the hidden dimension and the rank, we find that the performance drops significantly. On the other hand, LST is quite robust across a wide range of side network sizes.

We also consider another way to compare LoRA, Adapter and BitFit to LST in different memory budgets. While we drop $N \in \{0, 6, 12, 18\}$ layers in an interleaving manner to improve memory efficiency, we freeze the first $N$ layers and remove the corresponding inserted modules in other approaches. With this, other methods can achieve better memory efficiency because gradients do not propagate to those earlier frozen layers. We discuss the layer dropping and layer freezing with details in the following. In LST, we drop $\frac{N}{2}$ layers in both side encoder and side decoder. However, in other PETL approaches, we start from freezing layers in the encoder and then turn to freeze layers in the decoder (e.g. freezing 18 layers means freezing all encoder layers and first 6 decoder layers). We display the comparison in Figure 6, showing that LST has a better performance and memory trade-off and outperforms other methods in the low-memory regime. We also find that layer dropping generally reduces the training cost without hurting performance.

**The ablation of weight initialization on the side network.** We compare the different initialization strategies for the side network and demonstrate the results in Figure 7. "Random" denotes we randomly initialize the network while we use network pruning to select initial weights for the side network based on two importance measures, "Weight Magnitude" and "Fisher Information." In general, the initialization from the pruned network helps no matter the size of the side network, showing the effectiveness of our network pruning strategy.

**Comparison of LST to network compression methods and Side-tuning.** In Section 3.2, we mention that shortcut connections are added to every layer of the side network. We justify this design by comparing LST to two types of approaches: (1) network compression, which discards all shortcut connections and the entire backbone model; (2) Side-tuning, which only adds one shortcut connection to merge representations right before the output layer. Note that we do not drop any layer in the side network but only remove the shortcuts in this experiment. We also compare both distillation-based and pruning-based initialization methods as we describe in Section 2.2. We set the

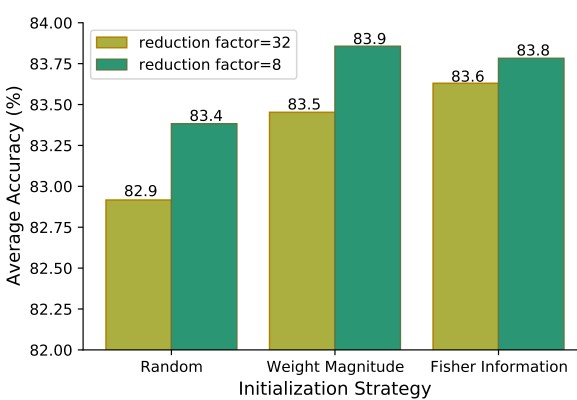

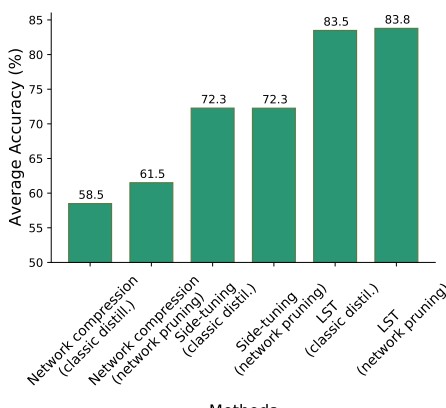

Figure 7: The ablation study on different initialization strategies. Y-axis denotes the average score over GLUE tasks.

Figure 8: The comparison between network compression, Side-Tuning, and LST on GLUE tasks.

reduction factor to 8 for all approaches. Figure 8 shows the comparison and LST outperforms the other two types of methods significantly. We conclude that PETL methods are stronger than network compression as they use the information from the backbone model. This also suggests that network compression approaches need to train more parameters to achieve the same level of performance as PETL methods. We also demonstrate the usefulness of intermediate shortcuts since LST surpasses Side-tuning by a large amount. Furthermore, we find that using distillation-based initialization or network pruning-based initialization provides similar accuracy in all setups. Note that our network pruning-based initialization method is more efficient since it does not involve training.

# 6 Conclusion

We propose Ladder Side-Tuning (LST), a parameter- and memory-efficient method to adapt a large backbone network. LST does not require backpropagation through the backbone network, which allows for significantly lower memory requirement during training than recently proposed parameter-efficient training techniques. We demonstrate that LST allows users to adapt a larger and more powerful backbone network to target tasks with a limited memory requirement, which cannot be achieved with recent parameter-efficient techniques. We also show that LST achieves a more efficient accuracy-memory trade-off than recent baselines, the impact of weight initialization of side networks, and the usefulness of intermediate shortcut connections. Finally, we show that the LST can be also extended beyond NLP tasks, with strong results on VL tasks. We hope that LST helps users with limited computational resources tune larger models in diverse domains.

## Acknowledgments

We thank the reviewers, Muqeeth Mohammed, Derek Tam, Prateek Yadav, and Gedas Bertasius for their helpful discussions. This work was supported by ARO Award W911NF2110220, ONR Grant N000141812871, and NSF-AI Engage Institute DRL-211263. The views, opinions, and/or findings contained in this article are those of the authors and not of the funding agency.

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
