# Supplementary Materials for
# LST: Ladder Side-Tuning for
# Parameter and Memory Efficient Transfer Learning

**Yi-Lin Sung**      **Jaemin Cho**      **Mohit Bansal**
UNC Chapel Hill
{ylsung,jmincho,mbansal}@cs.unc.edu

## 1   Comparison of different layer design in the ladder side network

As presented in Section 3.2, our side networks are built on Transformer blocks (same as the backbone network) and we use a gating mechanism to fuse the backbone information to the side network. Before choosing this specific architecture, we also explored several variations in choosing block modules and fusion methods. First, instead of using Transformer blocks for side network, we also test the performance of Adapter blocks (two-layer bottleneck structure), which show slightly smaller memory usage than Transformer blocks. Second, instead of using a gating mechanism to aggregate information from backbone and information from lower side network layers, we explore using cross-attentions (keys and values are the backbone network's representation while queries are the side network's representation) to achieve the same goal. Note that this design makes the side network have an additional attention layer compared to its backbone network counterpart. We demonstrate the results in Table 1. We find that our current side network design outperforms the others in terms of accuracy and has similar efficiency to adapter-based side network. We also observe that adding cross-attentions makes the training unstable and sensitive to the learning rates, making cross-attention have lower performance, even though it introduces heavy computation. Thus, we propose to use our current "Transformer block + gates" design for the ladder side tuning.

Table 1: Comparison of different designs of side network using the GLUE benchmark. We run each experiment for three seeds and report the average accuracy.

| Side Network Design | Update Param. per Task (%) | Memory Usage (GB) | Avg. Accuracy on GLUE (%) |
|---|---|---|---|
| Adapter block + gates | 2.07 | 6.5 | $83.1_{0.3}$ |
| Transformer block + cross attention | 2.68 | 10.4 | $83.0_{0.2}$ |
| Transformer block + gates (current design) | 2.29 | 7.0 | $83.8_{0.5}$ |

## 2   Hyper-parameters

We put the hyper-parameters used for NLP experiments (Table 1) in Table 2 and VL experiments (Table 3) in Table 3.

36th Conference on Neural Information Processing Systems (NeurIPS 2022).

Table 2: Hyper-parameters used for NLP experiments. Batch size is 100 for all methods.

| Method | Learning Rate | Other Hyper-parameters |
|---|---|---|
| Full fine-tuning | $3 \times 10^{-4}$ | - |
| Adapters | $3 \times 10^{-4}$ | hidden dimension=48 |
| LoRA | $3 \times 10^{-4}$ | rank=32 |
| BitFit | $3 \times 10^{-4}$ | - |
| Prompt-tuning | $3 \times 10^{-1}$ | number of prompts=100 |
| Ladder Side-Tuning | $3 \times 10^{-3}$ | r=8; index of layers are kept in side encoder and decoder=1,2,3,5,6,7,9,10,11 (drop 3 layers each) |
| Ladder Side-Tuning (T5-large) | $3 \times 10^{-3}$ | r=8; index of layers are kept in side encoder and decoder=1,3,5,7,9,11,13,15,17,19,21,23 (drop 12 layers each) |
| Ladder Side-Tuning (T5-3B) | $3 \times 10^{-3}$ | r=8; index of layers are kept in side encoder and decoder=23 (drop 23 layers each) |

Table 3: Hyper-parameters used for VL experiments. Batch size is 300 for all methods.

| Method | Learning Rate | Other Hyper-parameters |
|---|---|---|
| Full fine-tuning | $3 \times 10^{-4}$ | - |
| Adapters | $3 \times 10^{-4}$ | hidden dimension=192 |
| LoRA | $3 \times 10^{-4}$ | rank=150 |
| BitFit | $3 \times 10^{-3}$ | - |
| Prompt-tuning | $3 \times 10^{-3}$ | number of prompts=40 |
| Ladder Side-Tuning | $3 \times 10^{-3}$ | r=4; all layers in side networks are kept |