# OpenReview forum: "LST: Ladder Side-Tuning for Parameter and Memory Efficient Transfer Learning"
_NeurIPS.cc/2022/Conference — NeurIPS 2022 Accept_

### Official Review · Reviewer_L9hL · 2022-07-04

**Rating:** 3
**Confidence:** 4
**Soundness:** 2 fair
**Presentation:** 1 poor
**Contribution:** 2 fair

**Summary:**

This paper proposes Ladder Side-Tuning (LST), a method for parameter-efficient transfer learning with pre-trained neural networks. LST is an extension of the concept of Side-Tuning [1], which is a previous work for avoiding full parameter updates by adding small target side networks. In contrast to Side-Tuning, LST fuses the source and target models in the intermediate layer outputs by weighted sum and trains the model with several engineering techniques such as initialization of the weight by structural pruning and layer-wise dropout. Experiments confirm that LST can achieve competitive performances to several previous parameter-efficient transfer learning methods while largely reducing memory usage during the training time.

```
[1] Zhang, J. O., Sax, A., Zamir, A., Guibas, L., & Malik, J. (2020, August). Side-tuning: a baseline for network adaptation via additive side networks. In European Conference on Computer Vision (pp. 698-714). Springer, Cham.
```

**Questions:**

### Q1. Why is memory usage during training important?
The motivation explained in the Introduction is not clear to us. As far as we know, parameter-efficient transfer learning methods using large pre-trained models are motivated not only by the reduction of update parameters and memory usage but also by the prevention of overfitting [1,3,4]. There is also a focus on speeding up inference time for classical parameter reduction methods such as knowledge distillation [5] and pruning [6,7]. On the other hand, this paper focuses only on the limited issue of memory usage during training and does not consider the other important motivations enumerated above. We also consider that even the problem of memory usage has not been completely solved, since the experiment in Table 1 when using a huge pre-training network (T5-3B), ended up exceeding 16 GB, which is the standard for memory-saving environments in other evaluations. More to the point, in business, the training phase has relatively usable resources compared to inference, so providing an A6000 with the 48GB of memory used in the experiments in this paper for training is not much of a cost. Rather, LST or other methods adding parameters increase the cost of inference and cannot be used in resource-sensitive environments during inference. In summary, the research question for this paper is unclear and thus we are unable to determine the impact or accomplishment of the reported results. Please give us a specific practical scenario where LST would be useful.

```
[3] Haoming Jiang, Pengcheng He, Weizhu Chen, Xiaodong Liu, Jianfeng Gao, and Tuo Zhao. 2020. SMART: Robust and Efficient Fine-Tuning for Pre-trained Natural Language Models through Principled Regularized Optimization. In Proceedings of the 58th Annual Meeting of the Association for Computational Linguistics, pages 2177–2190, Online. Association for Computational Linguistics.
[4] Rabeeh Karimi Mahabadi, Sebastian Ruder, Mostafa Dehghani, and James Henderson. 2021. Parameter-efficient Multi-task Fine-tuning for Transformers via Shared Hypernetworks. In Proceedings of the 59th Annual Meeting of the Association for Computational Linguistics and the 11th International Joint Conference on Natural Language Processing (Volume 1: Long Papers), pages 565–576, Online. Association for Computational Linguistics.
[5] Hinton, Geoffrey, Oriol Vinyals, and Jeff Dean. "Distilling the knowledge in a neural network." arXiv preprint arXiv:1503.02531 2.7 (2015).
[6] Li, Hao, et al. "Pruning filters for efficient convnets." International Conference on Learning Representation (2017).
[7] Liu, Liyang, et al. "Group fisher pruning for practical network compression." International Conference on Machine Learning. PMLR, 2021.
```

### Q2. Why not compare LST to other parameter-efficient methods?
This is a related question to Q1.
One of the most popular methods for transfer learning with reduction of training parameters and amount of memory is distillation. A method particularly related to LST is FitNets[8], which minimizes the Kullback-Leibler divergence between the intermediate layer outputs of source and target models. The difference between LST and FitNets is whether the output of the source model is directly used as the output of the target model. In contrast to LST, the trained models with FitNets do not require the source model during inference time. Is LST more effective than simple and lightweight transfer learning methods like FitNets?
Furthermore, the more related study is LIT [2], a prior study that shares a source-target intermediate layer but also distillation. Is there any technical difference between LIT and LST other than the detailed learning techniques (initialization and Layer Dropout)?

```
[8] Romero, Adriana, et al. "Fitnets: Hints for thin deep nets." arXiv preprint arXiv:1412.6550 (2014).
```
### Q3. What is the definition of "network-pruning" in Fig. 8?
The definition of "network-pruning" in L301-L303 is ambiguous. In which strategy is this model pruned in Fig. 7? What is the reduction factor for pruning? Is the pruned model being fine-tuned?


**Limitations:**

Nothing to report.

**Strengths And Weaknesses:**

### Strength
  * Reducing memory usage during training time in comparison to Adapters and Prompt Tuning.

### Weakness
  * Novelty is limited since the idea of fusing intermediate layers is overlapped with an existing work (LIT, [2])
  * The motivation and research questions are unclear.
  * Lack of comparisons with baselines such as knowledge distillation, which are important for assessing the practicality of the proposed method.
  * Rather increasing memory usage during inference time.
  * Unclearness of experimental settings in particular the experiments of Fig.8.


LST is certainly effective in reducing training parameters and memory usage, but its practicality can not be judged based on the current contents because there is no comparison with more simple and lightweight methods such as linear probing (a transfer learning method updating only the target-task-specific layers) and knowledge distillation. Furthermore, the main idea of LST which is to fuse intermediate layers of source and target models is not novel since LIT [2] has already proposed it at ICML2019. Considering the above, we believe that this paper is not yet ready for publication because the weaknesses outweigh the strengths.

```
[2] Koratana, A., Kang, D., Bailis, P. &amp; Zaharia, M.. (2019). LIT: Learned Intermediate Representation Training for Model Compression. Proceedings of the 36th International Conference on Machine Learning, in Proceedings of Machine Learning Research 97:3509-3518 Available from https://proceedings.mlr.press/v97/koratana19a.html.
```

---

> ### Author Response · Authors · 2022-08-02
> **Response to Reviewer L9hL (part 1)**
>
> Thank you for your feedback. We address your concerns with the following replies.
>
> **(1) Novelty is limited since the idea of fusing intermediate layers is overlapped with an existing work (LIT, [2]) + The motivation and research questions are unclear.**:
>
> Our novelty is not just about adding the shortcuts from the backbone model to the side network. There are also the other two crucial techniques used in LST: layer dropping and network pruning. Without layer dropping, LST would not use such a small amount of memory in T5-large and T5-3B experiments (in Table 1 and L230 - 232). The network pruning technique used in LST is fast and simple (L127-129), providing better performance without many extra computational costs (in Figure 7).
>
> The core contributions of our paper are to analyze the memory requirement of current PETL methods (Sec 3.1) and to introduce LST, a parameter and memory efficient transfer learning method. Even though some of the techniques we used do not originate from us, we use them to present an important and novel goal of memory-efficient parameter-efficient transfer learning.
>
>
> **(2) Lack of comparisons with baselines such as knowledge distillation, which are important for assessing the practicality of the proposed method**:
>
> Recent language models (our backbone models) are usually trained on large-scale unlabeled datasets. To do knowledge distillation from the language model, we will need to do the training on the same scale of datasets. This step will take a significant amount of training time and many computational resources, and it violates our goal of efficiency. However, another approach is to do distillation via the downstream datasets. Therefore, for each task, we first apply distillation to make the side network (we remove all shortcuts) mimic the representation of the backbone network, and then we use this trained side network to initialize the weight in the Side-tuning. We use the objective proposed by LIT, and set the weight for KL divergence loss and intermediate presentation loss as 0.8 and 0.2. For distillation stage and fine-tuning stage, we search the optimal learning from \{3e-3, 3e-4\} and \{3e-3, 1e-3, 3e-4\}, respectively.
>
> The results are shown in the following table. We find there is a large gap between LIT’s and LST’s accuracy (and LIT is a bit better than Classic distillation [1]). This result aligns with our findings that using the intermediate representation during fine-tuning is useful (L 306-307), and supports our claims that the PETL method can achieve better performance than network compression approaches even when using a small number of trainable parameters (L 308-309).
>
> | | Avg GLUE score |
> | ----------- | :-----------: |
> | Classic distillation [1] | 55.9 |
> | LIT | 56.8 |
> | LST | **83.8** |
>
> [1] Hinton, G.E., Vinyals, O., & Dean, J. (2015). Distilling the Knowledge in a Neural Network. ArXiv, abs/1503.02531.
>
> > There is no comparison with more simple and lightweight methods such as linear probing (a transfer learning method updating only the target-task-specific layers) and knowledge distillation.
>
> Linear probing is a classic method to fine-tune pre-trained models in classification problems. However, note that the tasks we focus on in the paper are generation problems (the output is a sequence of tokens), where applying linear-probing is not trivial. In addition, since the language model head (linear output layer of decoder) costs around 30% of the whole language model parameters due to the large vocabulary size, fine-tuning the head is not parameter-efficient. Due to the above reasons, we didn’t compare LST to linear probing, but focused on comparing LST to other recent parameter-efficient approaches. We also report knowledge distillation results above.
>
> **(3) Rather increasing memory usage during inference time.**:
>
> We report the inference memory for all PETL methods in the following table. LST has an additional network, so it uses 15% more memory than full fine-tuning and Adapter when computing the attention in the side network. However, given that LST saves around 70% and 58% of memory that is used in full fine-tuning and Adapter, we believe the strength outweighs the weakness. Also, training a model costs much more memory than doing inference. Therefore, if we can train a model on a certain machine (e.g. edge devices), then we must be able to do inference on the same machine as well (if we consider the training and inference are done on the same machine).
>
>
> | |  Training memory (GB) | Inference memory (GB) | Avg. GLUE score |
> | ----------- | :-----------: | :-----------: | :-----------: |
> | Full fine-tuning | 17.6 | 1.95 | 85.2 |
> | Adapter | 13.0 | 1.96 | 85.3 |
> | LoRA | 12.6 | 1.95 | 85.3 |
> | BitFit     | 10.7 | 1.95 | 84.1 |
> | Prompt-tuning | *22.2 | 2.70 | 72.2 |
> | LST | **5.50** | 2.25 | 84.1|
>
> *Corrections from the current pdf: There was a typo in prompt-tuning training memory usage. We correct it from 11.6 to 22.2.

---

> ### Author Response · Authors · 2022-08-02
> **Response to Reviewer L9hL (part 2)**
>
>
>
> **(4) Q1. Why is memory usage during training important?**:
>
> > The motivation explained in the Introduction is not clear to us. As far as we know, parameter-efficient transfer learning methods using large pre-trained models are motivated not only by the reduction of update parameters and memory usage but also by the prevention of overfitting [1,3,4]. There is also a focus on speeding up inference time for classical parameter reduction methods such as knowledge distillation [5] and pruning [6,7]. On the other hand, this paper focuses only on the limited issue of memory usage during training and does not consider the other important motivations enumerated above.
>
> The purpose of PETL learning is to solve transfer learning with as few parameters as possible so that we can store a few parameters to the checkpoint for each task, and can totally eliminate the negative interference from other tasks [1, 2]. Knowledge distillation and pruning are usually not considered as PETL methods and are rarely compared in related literature [1, 2, 4] because they are not that parameter-efficient (performance drops significantly when parameter usage is smaller than 10% [3]) (L308 - 309) and they usually discard the backbone models (while PETL methods do not) (L119 - 122). On the other hand, in this paper, we comprehensively compare LST against recent PETL methods, such as Adapter, LoRA, BitFit and Prompt-tuning, and show the memory efficiency of our methods.
>
>
> [1] Houlsby, N., Giurgiu, A., Jastrzebski, S., Morrone, B., Laroussilhe, Q.D., Gesmundo, A., Attariyan, M., & Gelly, S. (2019). Parameter-Efficient Transfer Learning for NLP. ICML.
>
> [2] Mahabadi, R.K., Ruder, S., Dehghani, M., & Henderson, J. (2021). Parameter-efficient Multi-task Fine-tuning for Transformers via Shared Hypernetworks. ACL.
>
> [3] Frankle, J., & Carbin, M. (2019). The Lottery Ticket Hypothesis: Finding Sparse, Trainable Neural Networks. arXiv: Learning.
>
> [4] Davison, J. (2021). COMPACTER: Efficient Low-Rank Hypercomplex Adapter Layers.
>
>
> > We also consider that even the problem of memory usage has not been completely solved, since the experiment in Table 1 when using a huge pre-training network (T5-3B), ended up exceeding 16 GB, which is the standard for memory-saving environments in other evaluations.
>
> Note that we did include the results of T5-large + LST result which outperforms other PETL models by only using 13GB in Table 1. In our experiments, the other PETL methods could not be even used with T5-large within 16GB memory.
>
> > More to the point, in business, the training phase has relatively usable resources compared to inference, so providing an A6000 with the 48GB of memory used in the experiments in this paper for training is not much of a cost.
>
> We would like to point out that there are many academic labs that do not have A6000 or such large GPUs. Moreover, even in industrial production settings based on federated learning, training happens on many edge devices with small memory. LST can help to train larger models on edge devices as well.
>
> **(5) Q2. Why not compare LST to other parameter-efficient methods?**
>
> The purpose of distillation-based methods is different from that of LST. In our experiments, the performance of LIT, a distillation-based method, is behind our LST by a large margin under the similar parameter usage, as mentioned in (2). Please refer to our replies (2) and (4) for more details.
>
> **(6) Q3. What is the definition of "network-pruning" in Fig. 8?**
>
> > The definition of "network-pruning" in L301-L303 is ambiguous.
>
> Sorry for the confusion. We use network pruning both in initialization for the side network and Fig 8. The network pruning in Fig 8. means that we remove all shortcut connections and the entire backbone network (L303). More concretely, in this approach, we apply the network pruning method on the backbone model to extract the side network and only keep the side network and train it on the downstream tasks. This essentially becomes a typical network pruning procedure, so we called it “network pruning”. To avoid confusion, we will clarify the term we use to initialize the side network to “network pruning-based initialization.”
>
> > In which strategy is this model pruned in Fig. 7?
>
> We use the same pruning technique described in Sec. 3.3, and use the Fisher information as an importance score (L200).
>
> > What is the reduction factor for pruning? Is the pruned model being fine-tuned?
>
> The reduction factor is 8 (L228), which is the same as most experiments, and the pruned model is fine-tuned.

---

> > ### Comment · Reviewer_L9hL · 2022-08-08
> > **Replies to the initial response**
> >
> > Thanks for the detailed response and we are sorry it took so long to reply.
> > We understand that your arguments in your responses are as follows:
> >
> > 1. Contribution -- Proposing a new problem setting extending PETL, where the computation resources for training are restricted such as edge devices, by revealing that the prior methods consume much memory during training while they reduce trainable parameters.
> > 2. Novelty -- Applying existing techniques (side networks + layer dropping + pruning) to the new problem setting of PETL for reducing memory usage during training.
> >
> > In light of the above, the novelty of this paper lies primarily in its proposal of a new problem setting. We will additionally reply based on the above understandings.
> >
> >
> > ### About problem setting
> >
> > > The purpose of PETL learning is to solve transfer learning with as few parameters as possible so that we can store a few parameters to the checkpoint for each task, and can totally eliminate the negative interference from other tasks [1, 2].
> >
> > We think the problem in this paper is a bit different from the one addressed by prior PETL methods. As you answered, existing methods focus on solving various tasks with fewer parameters (parameter-efficiency) rather than reducing memory usage during training (memory-efficiency). On the other hand, the authors have extended this problem to add a new goal of memory-efficiency. If a paper poses a new problem setting, it should carefully situate that problem setting in logical and experimental comparison with related settings such as distillation, pruning, and other memory-efficient strategies. In this viewpoint, this paper fails to precisely position the problem setting because the problem is stated only in the context of PETL, and thus the reviewers pointed out the lack of comparison with several related studies (e.g., side-tuning and distillation). We believe that not only technical contributions but also contributions by proposing a new problem setting are equally important. However, we think that the current paper is insufficient in explaining this new problem setting, especially in terms of the following question.
> >
> > * What is the specific scenario that this problem is targeting and why is it so important?
> >   - More specifically, what are the concrete cases where increasing memory usage during inference is tolerated in exchange for reducing memory usage during training?
> >
> > To resolve this concern completely, we believe that the contents of the paper (e.g., introduction, related work, and experiments) need to be significantly revised. Thus, we do not intend to change our score in this time.

---

> > > ### Author Response · Authors · 2022-08-09
> > > **New reply and clarifications to L9hL**
> > >
> > > Thanks for the reply! We would like to clarify some important points. Please note that we have included the comparison of LST to network pruning (in Figure 8) and distillation (in our last reply part 1-(2)). Those compression methods are not comparable to LST because their usages are different from LST (L307-309): network compression (pruning and distillation) aims to produce a small model that can do fast inference on devices, whereas PETL methods aim to achieve similar performance as fine-tuning the whole model with updating as few parameters as possible. LST is still a PETL method as we keep and fine-tune the original language model (L121), and conduct experiments in the multi-tasking transfer learning setup (Section 4). The additional memory efficiency should be a benefit, but not a limitation either in applications or in motivation. We would like to re-emphasis the focus of this paper: **we introduce LST, a new memory-efficient PETL method, to help users with limited training memory settings to adapt larger backbone language models and achieve higher accuracy.** Since we already positioned our paper as a memory-efficient **PETL** method, we do not think our paper needs to be “significantly revised”. *We hope you agree with this response and increase your score.*
> > >
> > > Furthermore, we would like to claim that the additional inference memory usage of LST is negligible. In many real-world use cases, people often use batch size 1 for inference, making the model parameters the dominant part of memory consumption. As shown in the following table, in batch size=1, LST uses almost the same inference memory (0.88 GB) as the full fine-tuning (0.86 GB). Such a small increase in inference memory of LST should not be a major concern in practice.
> > >
> > >
> > > | |  Training memory (batch size=100) (GB) | Inference memory (batch size=100) (GB) | Inference memory (batch size=1) (GB) |
> > > | ----------- | :-----------: | :-----------: | :-----------: |
> > > | Full fine-tuning | 17.6 | 1.95 | 0.86 |
> > > | Adapter | 13.0 | 1.96 | 0.87 |
> > > | LoRA | 12.6 | 1.95 | 0.86 |
> > > | BitFit     | 10.7 | 1.95 | 0.86 |
> > > | Prompt-tuning | *22.2 | 2.70 | 0.87 |
> > > | LST | **5.50** | 2.25 | 0.88 |
> > >
> > >
> > >
> > > Moreover, LST opens up new possibilities for PETL methods for limited memory training scenarios, such as federated training and on-device training. In these setups, training is done in small devices, where the memory-efficiency of LST shines, which makes PETL can be applied in broader domains.

---

### Official Review · Reviewer_tNTu · 2022-07-09

**Rating:** 4
**Confidence:** 4
**Soundness:** 2 fair
**Presentation:** 3 good
**Contribution:** 2 fair

**Summary:**

This paper proposes Ladder Side-Tuning to improve the memory efficiency of transfer learning. The core idea is to freeze the backbone while adding an extra trainable side network. The side network takes multiple features extracted by the backbone and produces the output. This paper also shows that using the weight from the backbone to initialize the side network and layer dropping are helpful for memory-efficient transfer learning.

**Questions:**

See weaknesses.

**Limitations:**

Limitations are discussed in the appendix. Discussions about negative social impact are missing.

**Strengths And Weaknesses:**

Strengths:
1. Improving the memory efficiency of transfer learning is critical especially when we want to use large pre-trained models.
2. The proposed method is simple and can be applied to different model architectures.

Weaknesses:
1. The overall idea of the proposed method is very similar to Side-Tuning [1]. While the authors argue that the proposed method has a different goal and focuses on different tasks, I still find the overall technical novelty of the proposed methods is a bit limited.
2. In the related work section, I think the authors should add a section discussing the differences between the proposed method and previous memory-efficient transfer learning methods (e.g., [2,3]). In the experiment section, I think it is necessary to add direct comparisons between the proposed method and previous memory-efficient transfer learning methods.
3. There are some general memory-saving techniques for deep learning, such as reversible neural networks [4,5,6], and checkpointing [7]. I think it is better to add discussions about these techniques.
4. For the experiment section, I think the pre-trained model should be the same for all compared methods to ensure fair comparisons. To get results under larger memory constraints, the authors can increase the size of side networks.

[1] Side-Tuning: A Baseline for Network Adaptation via Additive Side Networks, ECCV 2020

[2] Tinytl: Reduce memory, not parameters for efficient on-device learning, NeurIPS 2020

[3] Rep-Net: Efficient On-Device Learning via Feature Reprogramming, CVPR 2022

[4] The Reversible Residual Network: Backpropagation Without Storing Activations, NeurIPS 2017

[5] Reformer: The Efficient Transformer, ICLR 2020

[6] Reversible Vision Transformers, CVPR 2022

[7] Training Deep Nets with Sublinear Memory Cost

---

> ### Author Response · Authors · 2022-08-02
> **Response to Reviewer tNTu**
>
> Thank you for your feedback. We address your concerns with the following replies.
>
> **(1) Ladder Side-tuning (LST) vs. Side tuning**:
>
> > 1. The overall idea of the proposed method is very similar to Side-Tuning [1]. While the authors argue that the proposed method has a different goal and focuses on different tasks, I still find the overall technical novelty of the proposed methods is a bit limited.
>
> We have explained the difference between LST and Side-tuning in L99-107, and in the following, we explain them in more detail.
>
> > -Motivation.
>
> As we mentioned in L101-103, the focus of LST is to reduce the training memory requirement of current PETL methods. In Sec. 3.1, we explain why current parameter-efficient methods are not very memory-efficient during training and suffer from high computational costs. On the other hand, the memory issue is barely discussed in the Side-tuning paper. This is also why we explore techniques such as layer dropping to further increase memory efficiency.
>
> > -Architecture designs.
>
> Besides both LST and Side-tuning are not restricted to specific architectures, we also mentioned in L104 - 106 that LST applies more advanced techniques to make it more efficient and powerful, such as shortcut connections, network pruning, and layer dropping. Without shortcut connections, the performance of LST will drop significantly as shown in Figure 8. Without layer dropping, LST would not use such a small amount of memory in T5-large and T5-3B experiments (in Table 1). Note that distillation of large language model of T5 is not trivial if you do not have large computational resources, since you need to access the large training data (e.g., C4). Instead, the network pruning technique used in LST doesn’t require lots of data, which is fast, simple, and meets our goal of efficiency.
>
> > -Applied tasks.
>
> Even though both LST and Side-tuning are applied to transfer learning tasks (adaptation tasks), LST focuses on modern Transformer architectures and different tasks (NLP and VL). Most experiments in Side-tuning are about vision tasks and CNNs, and we find simply applying Side-tuning to our scenario does not work well (Figure 8). Hence, we put effort into improving the accuracy and memory efficiency while adding minimal computational overhead in NLP and VL domains. Also, adapting large language models to cross-modal (e.g. VL) tasks has become a standard approach these years, and we believe LST brings contributions to this field by allowing researchers to make use of stronger and bigger models of its memory efficiency.
>
> **(2) Comparison to previous memory-efficient transfer learning methods**:
>
> > 2. In the related work section, I think the authors should add a section discussing the differences between the proposed method and previous memory-efficient transfer learning methods (e.g., [2,3]). In the experiment section,...
>
> Thanks for pointing out the missing related work [3]. However, we did already have a direct comparison to BitFit (tunes the bias terms), which is conceptually similar to both [2, 3], and showed the results in Table 1 (NLP experiments) and Appendix’s Table 2 (VL experiments). In general, LST performs similarly (NLP) or better (VL) than only tuning the bias terms, and uses less memory because no gradients are required for the backbone network. We will add the missing reference [2] in the revision.
>
> **(3) Discussion of general memory-saving techniques**:
> > 3. There are some general memory-saving techniques for deep learning, such as reversible neural networks [4,5,6], and checkpointing [7]. I think it is better to add discussions about these techniques.
>
> Thanks again for pointing out those related works. We will expand and include this discussion in our revision. We did not include the discussion about these works, since the memory saving from efficient base architectures and checkpointing is agnostic to the memory saving from LST. Researchers can combine those methods with LST if they pursue a higher level of memory efficiency.
>
>
> **(4) Pre-trained models should be same for all compared methods**:
> > 4. For the experiment section, I think the pre-trained model should be the same for all compared methods to ensure fair comparisons. …
>
> We believe that the comparison is still fair because all approaches are allowed to use the "same" memory budgets (L259 - 262). Researchers and practitioners would be interested in this comparison on the same memory budget setting since the current research trend is to use larger models to improve performance. LST would help those researchers and practitioners who aim to maximize performance under limited resources.

---

> > ### Comment · Reviewer_tNTu · 2022-08-08
> > **Thanks for the response**
> >
> > I appreciate the authors' response. I think using different pre-trained models makes it difficult to compare different methods. In addition, I think adding more extensive experiments justifying the value of the method when compared or combined with previous memory-saving techniques is important. Given current results, it is unclear whether the proposed method still has values when other memory-saving techniques are used. Therefore, I will maintain the original score.

---

> > > ### Author Response · Authors · 2022-08-09
> > > **New reply and clarification to reviewer tNTu**
> > >
> > > Thanks for the comments. Please see our clarifications below.
> > >
> > > >  I think using different pre-trained models makes it difficult to compare different methods
> > >
> > > As we mentioned in our previous response, this is a different type of **fair comparison** where we want to show our memory savings of LST can allow a bigger model to fit in the **same** memory budget (which is not possible to fit before); and even though we use larger pre-trained models, they still belong to the same architecture (T5 series), and hence, the performance gain is because of being able to fit a bigger model (in the same memory budget), not because of different network architectures. Overall, LST would help those researchers and practitioners who aim to maximize performance under limited resources.
> > >
> > >
> > > > In addition, I think adding more extensive experiments justifying the value of the method when compared or combined with previous memory-saving techniques is important.
> > >
> > > Please note that we have included the comparison of LST to some memory-saving techniques, such as network pruning (in Figure 8) and multiple distillation methods (in our reply to reviewer L9hL part 1-(2)).
> > >
> > > *We hope these replies address your final questions, and hopefully you can increase your score.*

---

### Official Review · Reviewer_qN8G · 2022-07-10

**Rating:** 4
**Confidence:** 4
**Soundness:** 3 good
**Presentation:** 3 good
**Contribution:** 2 fair

**Summary:**

This paper proposes a new parameter-efficient transfer learning (PETL) technique called Ladder Side-Tuning (LST). The proposed method trains a small and separate ladder side network that takes intermediate activations as input and output predictions by short-cut connections. During finetuning, the paper only requires backpropagation through the extra small ladder side network, and thus improves parameter and memory efficient transfer learning. The experiments on various models and learning tasks show the efficiency of the proposed method.

**Questions:**

(1) More discussions/analyses about the key differences between side-tuning and LST.
(2) More experiments show the claims/advantages that LST differs from side-tuning.


**Limitations:**

The authors adequately addressed the limitations and potential negative societal impact of their work.

**Strengths And Weaknesses:**

Strengths:
(1) The proposed LST only needs to compute backpropagation on small ladder side networks, which greatly improves the efficiency of PETL models.

(2) The experiments on various models and tasks (vision and language) show the efficiency of LST.

Weaknesses:

(1) The proposed LST shares a similar idea with side-tuning [53]. As claimed in side-tuning [53], “we propose a straightforward alternative: side-tuning. Side-tuning adapts a pre-trained network by training a lightweight “side” network that is fused with the (unchanged) pre-trained network via summation”. In Line 99-107, this paper claims LST there are differences in motivations, architecture designs, and applied tasks between [53] and LST. I summarize these three aspects as follows.

-Motivation. Side-tuning [53] focuses on incremental learning, an adaption task that extends n-category classification to (n+m)-category classification. The proposed LST aims to fine-tune large pre-trained models on downstream tasks, which is a general adaption task. Although the tasks are not extra the same, they are both adaptation tasks. Side-tuning [53] can be directly used to fine-tune large pre-trained models without major adjustment.

-Architecture designs. Side-tuning claimed that places no constraints on the structure of the base model or side network, allowing for the architecture and sizes to vary independently. It applied a residual on the top of the backbone in experiments. LST uses multiple layers as a side network of the backbone. I think the modification mainly depends on how large the gap between a source task and a target task. The main reason could be incremental learning has to maintain the source classification and only require a minor update. [53] used knowledge distillation for initialization and LST uses network pruning. I think both are available.

-Applied tasks. Side-tuning [53] focused on incremental learning and LST focuses on adaptation from a large pre-trained model to downstream tasks.

Therefore, compared with [53], the contribution of this paper is weak.

(2) In Line 295, the paper conducts some studies on w/ random and w/ network pruning. I wonder if knowledge distillation side-tuning also works, which is proposed by side-tuning[53]. As far as I know, both knowledge distillation and network pruning work in network compression. It would be nice if the paper studies why not directly use knowledge distillation as side-tuning does.

(3) In Line 301-309, the paper conducts some studies on side-tuning. I am interested in the analysis that we directly use side-tuning [53] for this task and simply use more intermediate layers.  Using more intermediate layers is widely used in deep neural networks, at least dating back to segmentation/classification/detection tasks in 2016.

---

> ### Author Response · Authors · 2022-08-02
> **Response to Reviewer qN8G (part 1)**
>
> Thank you for your feedback. We address your concerns with the following replies.
>
> **(1) Ladder Side-tuning (LST) vs. Side tuning**:
>
> We have explained the difference between LST and Side-tuning in L99 - 107, but we think there are some misunderstandings because your comments mean something different from what we meant/said in the paper.
>
> > -Motivation. Side-tuning [53] focuses on incremental learning… Side-tuning [53] can be directly used to fine-tune large pre-trained models without major adjustment.
>
> As we mentioned in L101-103, the focus of LST is to reduce the training memory requirement of current PETL methods. In Sec. 3.1, we explain why current parameter-efficient methods are not very memory-efficient during training and suffer from high computational costs. On the other hand, the memory issue is barely discussed in the Side-tuning paper. This is also why we explore techniques such as layer dropping to further increase memory efficiency.
>
> > -Architecture designs. Side-tuning claimed that places no constraints on the structure of the base model or side network,... I think both are available.
>
> Besides both LST and Side-tuning not being restricted to specific architectures, we also mentioned in L104 - 106 that LST applies more advanced techniques to make it more efficient and powerful, such as shortcut connections, network pruning, and layer dropping. Without shortcut connections, the performance of LST will drop significantly as shown in Figure 8. Without layer dropping, LST would not use such a small amount of memory in T5-large and T5-3B experiments (in Table 1). Note that distillation of the large language model of T5 is not trivial if you do not have large computational resources, since you need to access the large training data (e.g., C4). Instead, the network pruning technique used in LST doesn’t require lots of data, which is fast, simple, and meets our goal of efficiency.
>
> > -Applied tasks. Side-tuning [53] focused on incremental learning and LST focuses on adaptation from a large pre-trained model to downstream tasks.
>
> Even though both LST and Side-tuning are applied to transfer learning tasks (adaptation tasks), LST focuses on modern Transformer architectures and different tasks (NLP and VL). Most experiments in Side-tuning are about vision tasks and CNNs, and we find simply applying Side-tuning to our scenario does not work well (Figure 8). Hence, we put effort into improving the accuracy and memory efficiency while adding minimal computational overhead in NLP and VL domains. Also, adapting large language models to cross-modal (e.g. VL) tasks has become a standard approach these years, and we believe LST brings contributions to this field by allowing researchers to make use of stronger and bigger models of its memory efficiency.

---

> ### Author Response · Authors · 2022-08-02
> **Response to Reviewer qN8G (part 2)**
>
> **(2) Knowledge distillation + (3) Side tuning with more intermediate layers**:
> > (2) In Line 295, the paper conducts some studies on w/ random and w/ network pruning… It would be nice if the paper studies why not directly use knowledge distillation as side-tuning does.
>
> > (3) In Line 301-309, the paper conducts some studies on side-tuning. I am interested in the analysis that we directly use side-tuning [53] for this task and simply use more intermediate layers. ….
>
> As mentioned in the reply to the previous question, we didn’t use distillation because large language models are trained with large-scale datasets (e.g., C4), which makes distillation on the same dataset hard to conduct with limited resources, and ultimately violates our goal of ‘efficient training’.
>
> However, to address the concern, we still come up with another setting: distillation on the downstream datasets. Concretely, for each task, we first apply distillation to make the side network (we remove all shortcuts) mimic the representation of the backbone network, and then we use this trained side network to initialize the weight in the Side-tuning. We use the classic distillation method [1] to minimize the KL divergence between the output probability of the side network and the backbone network. For distillation stage and fine-tuning stage, we search the optimal learning from \{3e-3, 3e-4\} and \{3e-3, 1e-3, 3e-4\}, respectively.
>
> The following table shows that Side-tuning with distillation-based initialization (A, C) or network pruning-based initialization (B, D) provides similar accuracy on the GLUE benchmark in two setups: with and without intermediate connections. Note that our network pruning-based initialization method does not involve training.
>
>
> | | Side-network initialization | Intermediate connections | Avg GLUE score |
> | --- | :---: | :---:| :---: |
> | (A) Side-tuning | knowledge distillation | |  71.1 |
> | (B) | network pruning | | 72.0 |
> | (C) | knowledge distillation | ✔  | 83.4 |
> | (D) LST | network pruning | ✔  | **83.8** |
>
> - (A) Side-tuning w/ distillation-based initialization: We first train our side network by knowledge distillation on GLUE tasks, then fine-tune it on GLUE tasks.
> - (B) Side-tuning w/ network pruning-based initialization: We initialize the side network parameters via network pruning from the backbone network, then fine-tune it on GLUE tasks.
> - (C) Side-tuning w/ more connections: We first train our side network by knowledge distillation on GLUE tasks, then add intermediate connections and fine-tune it on GLUE tasks.
> - (D) Ladder Side-tuning (Ours): We initialize the side network parameters via network pruning from the backbone network, then add intermediate connections and fine-tune it on GLUE tasks.
>
> [1] Hinton, G.E., Vinyals, O., & Dean, J. (2015). Distilling the Knowledge in a Neural Network. ArXiv, abs/1503.02531.

---

### Official Review · Reviewer_yNyy · 2022-07-10

**Rating:** 6
**Confidence:** 4
**Soundness:** 3 good
**Presentation:** 4 excellent
**Contribution:** 3 good

**Summary:**

This paper proposes an efficient fine-tuning method for pre-trained language models. The main idea is to fix the models' weights,  calculate the layer-wise activation, project the activation to lower dimension, and feed the downsampled activation into a smaller side network. During training, only the side network's parameters are updated so that the memory cost is small. The authors propose an initialization method for the side network that directly takes weights from the pre-trained models. Experiments show that the proposed method can memory-efficiently train large language models while preserving decent accuracy.

**Questions:**

- I am a little bit confused why the initialization method could help. For example, the activation after layer 1 will be fed into the first layer of the side network, which is also initialized by the partial weights of layer 1 in the full model. With that being said, the activations seem to be processed by the same layer twice (or partially twice), and I am not sure whether this is a good choice if I had understood correctly. Although the ablation experiments show the comparison between different pruning criteria, I think it would be better to compare methods like "offsetting" or "random initialization".

- This paper emphasizes on training memory but I am curious about how would the inference memory of LST compare with other methods since inference memory may also be bottlenecks. For example, a device may be able to fit T5-base but not able to fit T5-large or T5-3B for inference. So even if we are able to train a T5-3B with smaller memory, it maybe also limit the practical usage. This is somehow related to the comparison in Table 1 where the authors train a larger network for comparison.

- The experiments on LoRA seem to use a larger rank that most works suggested (for example most works use rank smaller than 16 which will lead to smaller than 1% trainable parameters), and original LoRA paper stated that they can save more than 2/3 VRAM on GPT-3 while in this paper the memory saving is fewer.

**Strengths And Weaknesses:**

- The memory saving is significant compared to other methods due to the extracted layer-wise representation and smaller side network.
- The number of parameters is smaller than full finetuning which saves the cost of storing parameters.
- Propose an initialization method for side networks that uses Fisher information and structured pruning.

---

> ### Author Response · Authors · 2022-08-02
> **Response to Reviewer yNyy**
>
> Thank you for your feedback. We address your concerns with the following replies.
>
> **(1) Weight initialization**:
> > I am a little bit confused why the initialization method could help. …, I think it would be better to compare methods like offsetting'' or random initialization''.
>
> Please note that in Figure 7, we did show the result with random initialization. In addition, we also tried to shift the side network 1 layer ahead, meaning that the activation after layer 1 is fed into the 2nd layer of the side network. This design has a better intuition that the activation from the backbone model would not be processed by similar weights twice. However, we found this design performed a bit worse than our current version, and we choose to only present the better and current version of LST. Another explanation would be that the backbone activation is fed into a learnable downsampling layer before feeding to the side network, so the representation will not be the same representation that has already been processed.
>
>
> **(2) Training and Inference memory**:
> > This paper emphasizes on training memory but I am curious about how would the inference memory of LST compare with other methods … This is somehow related to the comparison in Table 1 where the authors train a larger network for comparison.
>
> We report the inference memory for all PETL methods in the table below. LST has an additional network, so it uses 15% more memory than full fine-tuning and Adapter when computing the attention in the side network. However, given that LST saves around 70% and 58% of memory that is used in full fine-tuning and Adapter, we believe the strength outweighs the weakness. Also, training a model costs much more memory than doing inference.
> Therefore, if we can train a model on a certain machine (e.g. edge devices), then we must be able to do inference on the same machine as well (if we consider the training and inference are done on the same machine).
>
>
> | |  Training memory (GB) | Inference memory (GB) | Avg. GLUE score |
> | ----------- | :-----------: | :-----------: | :-----------: |
> | Full fine-tuning | 17.6 | 1.95 | 85.2 |
> | Adapter | 13.0 | 1.96 | 85.3 |
> | LoRA | 12.6 | 1.95 | 85.3 |
> | BitFit     | 10.7 | 1.95 | 84.1 |
> | Prompt-tuning | *22.2 | 2.70 | 72.2 |
> | LST | **5.50** | 2.25 | 84.1|
>
> *Corrections from the current pdf: There was a typo in prompt-tuning training memory usage. We correct it from 11.6 to 22.2.
>
>
>
> **(3) Low memory usage of LoRA**:
> > The experiments on LoRA seem to use a larger rank that most works suggested (for example most works use rank smaller than 16 which will lead to smaller than 1% trainable parameters),
>
> In Figure 5, we have shown the results of LoRA with different ranks (4, 8, 16, 32). Using a smaller rank indeed helps LoRA to use less parameters, but the memory cost still remains similar. However, LST can attain both parameter-efficiency and memory-efficiency simultaneously.
>
> > and original LoRA paper stated that they can save more than 2/3 VRAM on GPT-3 while in this paper the memory saving is fewer.
>
> Although the LoRA paper stated they can reduce the memory by not saving the Adam optimizer states for frozen parameters, all PETL approaches, including LST, benefit from such memory saving via optimizer engineering (e.g., by simply filtering optimizer parameters with `requires_grad=False` in case of PyTorch). In other words, LST would also save more than ⅔ of VRAM on GPT-3 with the same technique. However, LoRA doesn’t consider the training cost of the intermediate activations and their gradients, which are the dominant terms in training memory cost.

---

> > ### Comment · Reviewer_yNyy · 2022-08-07
> > **Thank you for your response**
> >
> > I would thank the authors for addressing my concerns. I have no additional question at this time.

---

> > > ### Author Response · Authors · 2022-08-09
> > > **New reply to reviewer yNyy**
> > >
> > > We are glad that our response addressed your concerns. Since there are no additional questions, it would be great if you can consider increasing your score or feel free to ask more questions. Thank you for the discussion.

---

### Author Response · Authors · 2022-08-02
**General Response**

We thank the reviewers for their feedback. We appreciate that reviewers agree with our contribution to reducing memory costs for PETL methods (all reviewers), consider our approach simple (reviewer tNTu), general (reviewer tNTu), and effective (reviewer qN8G), and agree with our technical contribution (reviewer yNyy).

We conducted new experiments to address some reviewers’ concerns, and we will add them in the revision.
* LST vs. Side-tuning + distillation [reviewer qN8G]
* LST vs. Side-tuning + distillation + shortcuts [reviewer qN8G]
* LST vs LIT [reviewer L9hL]
* Inference memory of LST [reviewer yNyy and reviewer L9hL]

We hope our replies can address the concerns, and please let us know if there are any new questions.

---

### Author Response · Authors · 2022-08-07
**Gentle reminder for author-reviewer discussion (ends Aug 9)**

Dear Reviewers,

This is a gentle reminder that the author-reviewer discussion ends on Aug 9 (Next Tuesday).  We appreciate your useful comments. We have provided detailed responses to them and hope that we have addressed your suggestions/questions. We are looking forward to hearing back from you about our response and also whether we can answer any remaining questions from you before the discussion period ends. Thank you!

Best,

Authors

---

### Meta-Review · Area_Chair_r4X5 · 2022-08-26

**Recommendation:** Accept
**Confidence:** Less certain

**Metareview:**

The paper contributes a novel training methodology based on a ladder network for a scenario of fine-tuning under limited memory constraints.

Two out of three reviewers recommended rejecting the paper. Reviewer L9hL noted that the paper is related to distillation or network prunning and is not discussed sufficiently. The method is related but not the same as distillation or network prunning. Both of these techniques require an additional computation step. I agree with Reviewer tNTu that it would strengthen the paper to make a one-to-one comparison with representative distillation or network prunning methods, but I also feel it is not strictly necessary as they have (slightly) different use cases. Hence, I feel it is more of an issue with writing, than with the core contribution.

It was also brought up that Side Tuning is a similar method from '19. My understanding is that Figure 8 and the rebuttal properly discusses this point and shows that the design choices made by the Authors (e.g. injecting the representation at multiple stages) are crucial innovations, and do not diminish novelty in my opinion.

All in all, I believe this is a solid contribution that is another tool that will help democratize large-scale models. It is my pleasure to recommend acceptance. Please remember to address reviewers' remark, and please pay special attention to better contextualizing your work in the broader field of memory efficient training of neural networks.


**Award:**

No

---

### Decision · Program_Chairs · 2022-09-14

Accept